# Effect of disclosing the relation between effort and unit reliability on system reliability: An economic experiment

**Ryoji Makino[1], Kenju Akai [2]\*, Jun-ichi Takeshita[1], Takanori Kudo[3], Keiko Aoki[4]**

**1** National Institute of Advanced Industrial Science and Technology, Onogawa, Tsukuba, Ibaraki, Japan,
**2** Shimane University, Izumo, Shimane, Japan, **3** Setsunan University, Neyagawa, Osaka, Japan, **4** Kyushu University, Nishi-ku, Fukuoka, Japan

\* akai@med.shimane-u.ac.jp

## Abstract

The purpose is to experimentally examine the effect of disclosing the risk probability of each unit in a production system on human behavior and the resulting system reliability. We used an economic experiment based on the theoretical model of Hausken (2002) to evaluate the effect of disclosing the relation between effort and unit reliability. We conducted first the non-disclosed-risk experiment and then the disclosed-risk experiment within subjects in both series and parallel systems. Our experimental results show that disclosing the relation between effort and unit reliability has two positive effects. First, subjects succeeded in improving the system reliability while cutting back on efforts to reduce the risk of their units when the risk probability was disclosed. In each system, the disclosed-risk condition achieves significantly higher system reliability on average than does the non-disclosed-risk condition, although the average level of effort is significantly lower under the disclosed-risk condition than under the non-disclosed-risk condition. Second, disclosing the risk probability simplified the subjects' decision-making process and reduced its cost because subjects made their decisions on the amount of effort to exert based only on the risk probability information without considering other factors, such as the number of accidents.

## 1. Introduction

The role of human behavior and decision-making in the design and operation of engineering systems, including those in the chemical, aviation, nuclear, health care, and construction industries, is crucial [1]. Therefore, it has been argued that human behavior should be considered in probabilistic risk analysis (PRA) to assess system reliability [2]. However, PRA has primarily taken a non-behavioral, physical engineering approach to estimating risk and assessing the reliability of a system [3], mainly because of the difficulty in understanding human behavior.

Hausken (2002) presented a way to deal with this issue by integrating game theory into PRA [3]. He introduced models that express production systems in which workers are equipped with units that are connected in series (called a "series system") and in parallel (called

**Funding:** This study was supported by JSPS KAKENHI Grant Number 15K01237 (Recipient: Ryoji Makino) and Leading Initiative for Excellent Young Researchers, MEXT, Japan (Recipient: Kenju Akai).

**Competing interests:** The authors have declared that no competing interests exist.

a "parallel system"), and he analyzed the situation in which (i) system reliability depends on people's intentional efforts to reduce risk and (ii) people influence each other in deciding the degree of effort. In particular, he focused on the free-rider problem that emerges when expending resources to reduce risk and demonstrated that when players place different values on system reliability, a conflict of resource allocation arises. His study aimed to merge the behavioral theory of conflict with the physical world to produce more accurate estimates of system reliability. Hausken's (2002) work was significant because it provided the first organized linkage between reliability analysis based on PRA and game theory in which system reliability was treated as a public good [4].

Integrated models of PRA and game theory are important for a number of reasons. First, such models take into account the intentional behavior of people rather than random events or acts of nature, which can easily be assessed by ordinary PRA models. Second, they can serve as base models for the application of mechanism design theory to design a system that reduces risk [5]. In addition, since the work of Hausken (2002), a number of theoretical and simulation-based studies of PRA and game theory have been presented in the field of anti-terrorism policies [6–8], so these models seem to have useful real-world applications. The original idea of Hausken who refer to Harshlierfer's idea of home and security. The series system is for the dike to protect an island from floods and the parallel system is for anti-missile battery that each citizen maintains to prevent missile. Our idea is also based on such a home and security so that we show the future issues applying for hazard map for disasters.

A number of previous studies, including Hausken (2000), have theoretically analyzed the models with complete information and not dealt with those with incomplete information. In that context, it is assumed that the risk probability is disclosed to the players of the game model and that they know the relationship between their strategies and the resulting equilibrium risk level.

Thus, in the context of an integrated model of PRA and game theory, this study aims to experimentally examine the effect of disclosing the risk probability of each unit in a production system on human behavior and on the resulting reliability of the production system using human subjects in a laboratory. An economic experiment based on the theoretical model of Hausken (2002) was employed to achieve this goal. Following Hausken (2002), we assume that the player has the same risk in each unit, that is, symmetric risk condition.

To evaluate the effect of disclosing the relation between effort and unit reliability, we conducted one experiment in which the risk probability was disclosed to subjects (hereafter, "the disclosed-risk condition") and one in which the risk probability was not disclosed (hereafter, "the non-disclosed-risk condition"). Comparing the results of these experiments, we can understand the effect of disclosing the relation between effort and unit reliability, and we focus on understanding how subjects' behavior and the system's reliability are affected by disclosing the relation between effort and unit reliability.

This analysis is related to the further development suggested by Hausken (2002), which is a case of games with incomplete information. Theoretical literature of incomplete information related to this study is Bier, Oliveros, Samuelson (2007) and Hausken (2014) [9, 10]. Both of them analyze the situation in which defenders have information asymmetry for attackers in the home and security problem. However, as our best of knowledge, there is no experiment which shows the impact of disclosing probability of risks. Therefore, it is not clear what will happen to player's behavior and system reliability when information about risk is incomplete. This is the first study that has analyzed the integrated model of PRA and game theory using an economic experiment.

Additionally, the theoretical model is calculated in the one-shot game in which the players only play the game once. We extend this situation into the repeated game to see the interaction

effect between players. We statistically analyze what happens after they experience the accident or know the counterpart's effort. We believe this experiment is worthwhile because it could shed new light on how the disclosure of risk probability affects the level of system reliability.

The structure of the rest of this paper is as follows. The next section describes the model developed by Hausken (2002) and makes the game theoretic prediction of subjects' behavior in our experiment. Section 0 presents the experimental design and procedure. We show the experimental results and provide discussions in sections 0 and 0. Section 0 concludes the article.

## 2. Theoretical model

### 2.1 Description and notations

The design of our experiment is based on the theoretical model presented by Hausken (2002) to study the system reliability of a series, parallel, summation, or combined system. A system can represent, for example, a factory consisting of some machine units connected in various ways. In our experiment, we examine a series system and a parallel system, each consisting of two units, which are the most fundamental classes. This subsection both explains our experimental setting and also serves as a brief and simplified description of Hausken's (2002) model.

Fig 1 depicts the structure of a series system. It is assumed that unit $i$ (= 1, 2) is equipped with maintenance worker $i$, so there are two workers in a system. Hausken (2002) provided examples of a series system showing (i) a chain that is not stronger than its weakest link and (ii) a case of flood protection on an island presented by Hirshleifer [11, 12].

The reliability of unit $i$ in period $t$, which is the probability of unit $i$ continuing to function in period $t$ = 0, 1,,, T, depends on maintenance worker $i$'s behavior in period $t$. Maintenance worker $i$ can increase the reliability of his/her unit in period $t$, $P_{i,t}$ by making efforts $e_{i,t}$ to maintain it. We assume maintenance workers have to incur costs to make efforts. The relationship between the reliability of unit $i$ and the effort level of worker $i$ in period $t$ is expressed as $P_{i,t} = p(e_{i,t})$, $p'(e_{i,t}) > 0$. In other words, unit $i$ breaks down with probability $1 - p(e_{i,t})$when worker $i$'s effort level is $e_{i,t}$. Note that an accident occurs with probability $1 - p(e_{i,t})$ in each period t.

The reliability of an entire series system depends on the reliability of each unit. Since a series system functions when both units function, the reliability of such a system in period $t$, $P_{series,t}$, is given by $P_{series,t} = p(e_{1,t}) * p(e_{2,t})$.

The subjects in our experiment play the role of maintenance workers, and they decide how much effort to exert. Depending on their effort levels, the reliability of their system varies. An accident (one that is, needless to say, hypothetical in our experiment) happens with probability

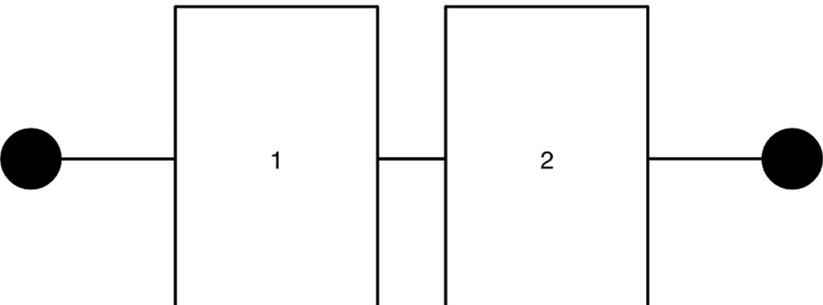

**Fig 1. Series system.**

**Table 1. Relation between effort and unit reliability.**

| Effort $e_{i,t}$ | 0 | 1 | 2 | 3 | 4 | 5 | 6 | 7 | 8 | 9 | 10 |
|---|---|---|---|---|---|---|---|---|---|---|---|
| Unit reliability $p(e_{i,t})$ | 0.20 | 0.20 | 0.60 | 0.60 | 0.60 | 0.60 | 0.90 | 0.90 | 0.90 | 0.90 | 0.90 |
| Effort $e_{i,t}$ | – | 11 | 12 | 13 | 14 | 15 | 16 | 17 | 18 | 19 | 20 |
| Unit reliability $p(e_{i,t})$ | – | 0.90 | 0.95 | 0.95 | 0.95 | 0.95 | 0.95 | 0.95 | 0.95 | 0.99 | 0.99 |

$1 - p(e_{1,t}) * p(e_{2,t})$. As explained later, if an accident happens during one period in the experiment, subjects do not receive a reward for that period.

In our experiment, efforts to maintain the machine unit are represented by the payment of "tokens" out of an endowment provided by the experimenter at the beginning of a period. The actual relationship between the effort level and the unit reliability applied in the experiment is shown in Table 1 and is the same in all sessions.

Subjects face a trade-off between the reliability of their unit and the cost of their efforts. They can reduce the risk of not being paid due to accidents by making efforts to maintain their units. However, in order to do that, they have to incur costs.

Furthermore, subjects face not only the above trade-off but also interdependence with their co-workers. It might be optimal for a worker to make little effort if his/her co-worker makes enough effort. In our experiment, subjects must decide their own effort levels before observing their co-workers' effort levels, which means they play simultaneous-move games.

Fig 2 depicts the structure of a parallel system. The analysis of a parallel system is the same as that of a series system except for the system reliability. The reliability of a parallel system is expressed as $P_{parallel.t} = 1 - (1 - p(e_{i,t}))(1 - p(e_{i,t}))$. Note that this type of system breaks down only if both units break down simultaneously, so it is, in general, more robust than a series system is. As Hausken (2002) showed, subjects in a parallel system are expected to make less effort than those in a series system because they have an incentive to free ride on their co-worker's effort. This free-rider behavior is driven by the fact that the reliability of a parallel system remains high when at least one maintenance worker makes enough effort and the reliability of his/her machine unit is high.

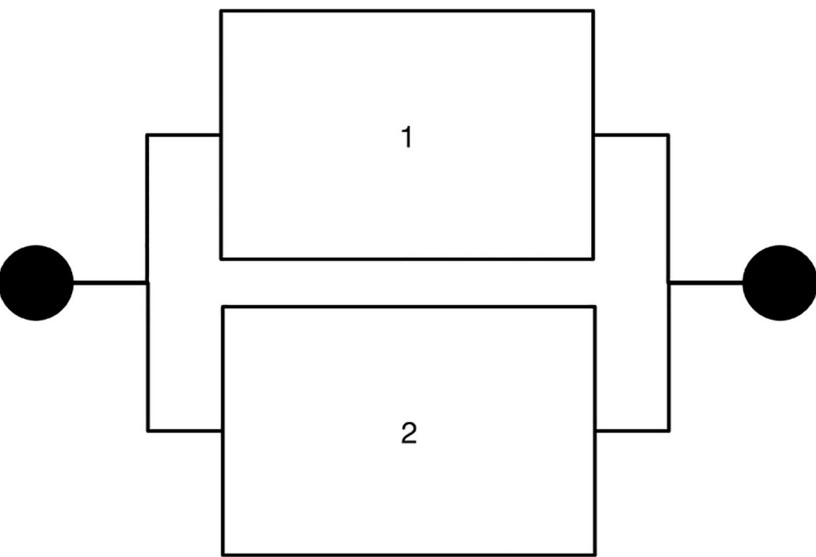

**Fig 2. Parallel system.**

## 2.2 Parameter settings

First, we describe the payoffs in the series system. In each period, each of the two subjects ($i$ = 1, 2) in a group receives an endowment of 20 tokens. Subjects can either keep these tokens for themselves or pay $e_{i,t}$ tokens ($0 \leq e_{i,t} \leq 20$) to increase the unit reliability. Their decisions about $e_{i,t}$ are made simultaneously. Depending on $e_{1,t}$ and $e_{2,t}$, the reliability of the groups' series system is computed according to the formula $p(e_{1,t}) * p(e_{2,t})$. When an (hypothetical) accident does not happen, each subject in each group receives 25 tokens. This payment represents the wage for their maintenance work and can be interpreted as a scaling factor $b$ of Hausken's model. An accident happens with a risk probability of $1 - p(e_{1,t}) * p(e_{2,t})$, and, in that case, the subjects in the group receive nothing. The expected payoff for each subject $i$ in period $t$ is given by

$$\pi^e_{series,i,t} = 20 - e_{i,t} + 25 \prod_{i=1}^{2} p(e_{i,t}).$$

The total payoff that subjects earn in the non-disclosed-risk (disclosed-risk) condition of the series system is the sum of the period payoffs over the first (last) 30 periods of the session, as explained in section 0.

Second, in the parallel system, only the payoff function for the subjects differs from the series system. The expected payoff in each period for each subject $i$ is now expressed as follows:

$$\pi^e_{parallel,i,t} = 20 - e_{i,t} + 25\{1 - \prod_{i=1}^{2}[1 - p(e_{i,t})]\}.$$

## 2.3 Nash equilibrium

Each player maximizes his/her expected utility in each period given that another player is also maximizing his/her expected utility. The equilibrium concept widely used in economic analysis is the Nash equilibrium, which is defined as the strategy profile from which no player has an incentive to deviate if the other players do not deviate [13]. In the series system, a strategy profile $e^*_t = (e^*_{1,t}, e^*_{2,t})$, where the values in the brackets stand for players' effort levels measured by the number of tokens, is a Nash equilibrium if

$$20 - e^*_{1,t} + 25p(e^*_{1,t})p(e^*_{2,t}) \geq 20 - e_{1,t} + 25p(e_{1,t})p(e^*_{2,t})$$

$$20 - e^*_{2,t} + 25p(e^*_{1,t})p(e^*_{2,t}) \geq 20 - e_{2,t} + 25p(e^*_{1,t})p(e_{2,t})$$

$$\forall e_{i,t} \in \{1, 2, \ldots, 20\}, (i = 1, 2).$$

In the parallel system, a Nash equilibrium $(e^{**}_{1,t}, e^{**}_{2,t})$ is expressed as follows:

$$20 - e^{**}_{1,t} + 25\{1 - [1 - p(e^{**}_{1,t})][1 - p(e^{**}_{2,t})]\} \geq 20 - e_{1,t} + 25\{1 - [1 - p(e_{1,t})][1 - p(e^{**}_{2,t})]\}$$

$$20 - e^{**}_{2,t} + 25\{1 - [1 - p(e^{**}_{1,t})][1 - p(e^{**}_{2,t})]\} \geq 20 - e_{2,t} + 25\{1 - [1 - p(e^{**}_{1,t})][1 - p(e_{2,t})]\}$$

$$\forall e_{i,t} \in \{1, 2, \ldots, 20\}, (i = 1, 2).$$

In our experimental setting, especially when the relationship between the reliability of unit $i$ and the effort level of worker $i$ shown in Table 1 is disclosed to the subjects, the Nash equilibria

are (0, 0) and (6, 6) for the series system and (0, 6), (6, 0), and (3, 3) for the parallel system. The Nash equilibrium in the series system implies that subjects pay nothing or pay all of their efforts so that the accidents will frequently occur. On the other hand, the Nash equilibrium in the parallel system implies that one of the subjects will become a free rider but the frequency of accidents will be decreased.

## 3. Experimental design and procedures

### 3.1 Design

Our experiment consists of four treatment conditions, as presented in Table 2. These conditions include a series system with and without the disclosure of the risk probability of a unit (the disclosed-risk condition and the non-disclosed-risk condition, respectively) and a parallel system with and without this disclosure. There are six experimental sessions altogether. Sessions 1, 2, and 3 are for the series system and sessions 4, 5, and 6 are for the parallel system.

We employ within-subject design to evaluate the difference in the behavior between the system with and without the disclosure of the risk probability of a unit. Each session consists of 60 periods. In the first 30 periods, subjects play the game under the non-disclosed-risk condition, which means that they do not know the quantitative relationship between their effort level and the reliability of their unit. As mentioned earlier, they know that the unit reliability is a non-decreasing function of the effort level. In the last 30 periods, the subjects play the same game under the disclosed-risk condition, which means that they do understand the relationship between their effort level and the reliability of their unit.

The total number of subjects in a session is $N = 12$, and the subjects are randomly partitioned into six groups of size $n = 2$ in each of the 60 periods. Thus, the group composition is randomly changed from period to period. In a given session, the same $N$ subjects play 30 periods under the non-disclosed-risk condition and 30 periods under the disclosed-risk condition.

The model proposed by Hausken (2002) assumes complete information, which means the players know the specific functional form of $p(e_{i,t})$. However, in a real work situation, reliability (in other words, the risk level) is rarely known. Accordingly, we explore experimentally how subjects' decision-making is affected and how the resulting system reliability varies when the risk probability is not disclosed. This experiment is expected to shed new light on the role of disclosing the relation between effort and unit reliability in enhancing the reliability of various systems.

### 3.2 Procedures

The experiment was conducted at Niigata University. The subjects were students from various faculties, including economics. Subjects were recruited through a flyer and voluntarily registered their names and email addresses on the website. When registering, they had to agree to a consent form explaining the freedom to opt out, the rewards to be gained from the results of

**Table 2. Treatment conditions.**

|  | Sessions 1–3: Series system 60 periods in each session | Sessions 4–6: Parallel system 60 periods in each session |
|---|---|---|
| Non-disclosed-risk condition (First 30 periods) | • 12 subjects in each session<br>• Six groups of size n = 2 in each period<br>• Random group composition in each period | • 12 subjects in each session<br>• Six groups of size $n = 2$ in each period<br>• Random group composition in each period |
| Disclosed-risk condition (Last 30 periods) | Conditions are the same as the non-disclosed-risk condition other than the disclosure of the risk probability | Conditions are the same as the non-disclosed-risk condition other than the disclosure of the risk probability |

the experiment, and the anonymity of the data. A paper consent form was distributed and read to all subjects at the beginning of the experiment to reaffirm the contents of the consent form. Subjects had to agree to the content and sign and submit the consent form in order to participate in the experiment.

The data collection complied with the Law on the Protection of Personal Information in Japan. All institutions and universities to which the author belongs did not require ethical approval for science research, except in instances that could be deemed life-threatening or harmful to human subjects. Subjects were not informed of the identities of other group members. Each subject will be allowed to participate in only one session.

Although the series and parallel systems follow the same procedure, they use different formulas to compute system reliability. The experiment is conducted in a computerized laboratory where subjects interact anonymously with each other. To conduct the experiment, we use the experimental software "z-Tree" developed by Fischbacher who defines it as "*The z-Tree software is implemented as a client-server application with a server application for the experimenter, called z-Tree, and a client application for the subjects, called z-Leaf. The applications are programmed in C++ (Visual C++ 2015, MFC) and run on all recent released x86 32 bit and 64 bit versions of Windows, starting with Windows XP SP3.*" [14]

The experimental operation process is as follows:

*Step1. (Deciding subjects)*: More than twelve subjects including two or three extras wait in the lobby before the session starts. Then, they are selected by the lottery to be decide who join the actual session. The selected twelve subjects are guided to the experimental room which has twelve laptops. The person who failed by the lottery receive 500 Japanese Yen (about 4.20 USD at that time) to leave this session. They have a chance to join the other session.

*Step2. (Reading consent form)*: The experimenter read aloud the consent form and the subjects hand it to the experimenter if they agree with it.

*Step3. (Reading instruction)*: The non-disclosed-risk condition starts. The subjects listen to the recorded instruction and read it silently by themselves at the same time. The instruction explains that the experiment has two parts and that each part has 30 periods. Then, the actions that subjects can take in the experiment and the method for computing the reward that subjects receive at the end of the experiment are presented. Then, they are given 7 minutes to review the instruction and make a strategy.

*Step4. (Decision making)*: At the beginning of the first period, they are randomly divided into six groups of two subjects by the computer. Each subject receives 20 tokens as his/her endowment. They decide how many tokens they wish to pay to maintain the machine unit before observing the number of tokens paid by their counterpart and without knowing the functional form of $p(e_{i,t})$ shown in Table 1.

*Step5. (Simulating accident)*: According to the number of tokens paid, the reliability of each machine unit and, hence, the system reliability of the group is computed. The occurrence of an accident is simulated based on the computed system reliability.

*Step6. (Calculating payoffs)*: The result of accident simulation is revealed to the subjects. When an accident happens, the payoffs in that period for the subjects in the group are determined as $\pi_{i,t} = 20 - e_{i,t}$. When an accident does not happen, the payoffs are $\pi_{i,t} = 20 - e_{i,t} + 25$.

*Step7. (Revealing the counterpart's effort)*: The subjects can see the number of tokens paid by their counterpart, whether or not an accident happened, and his/her own payoffs in the period. That is the final step in the first period.

*Step8 (Repeating the periods)*: After the first period completes, the second period begins. Groups are randomly changed, each subject receives another endowment of 20 tokens, and the process continues as described in the above seven steps. These procedures are conducted in each period until the 30th period ends. Subject *i*'s total payoff for the first 30 periods (i.e., the non-disclosed-risk condition) is computed as $\sum_{t=1}^{30} \pi_{i,t}$.

*Step9. (Revealing the probability)*: The disclosed-risk condition starts. At the beginning of the 31st period, the experimenter provides Table 1 to the subjects and explain that the experimental procedure is the same as that of the first 30 periods except that they can see the risk of accident.

*Step10. (Repeating the periods)*: Same six steps, Steps 3 to 8, in the non-disclosure condition repeated from 31 to 60th periods. After the 60th period ends, subject *i*'s total payoff for the last 30 periods is computed as $\sum_{t=31}^{60} \pi_{i,t}$.

*Step11. (Paying rewards)*: The reward actually paid to a subject is a random selection of either the total payoff from the non-disclosed-risk condition or that from the disclosed-risk condition. An exchange rate of 1 token = 3 Japanese Yen is applied.

## 4. Experimental results

### 4.1 Subject

We have observations from 36 subjects for the series system and the parallel system, respectively. Sessions 1 and 4 were held in November 2013, and the other sessions were held in January 2014. Each experimental session lasted about two hours, and subjects earned, on average, 4,228 Japanese Yen (about US $40 at the time), including a participation fee of 1,500 Japanese Yen. Table 3 shows the gender, average rewards and extra subjects who failed by the lottery in each session.

### 4.2 Series system

**4.2.1 Contributions.** The average number of tokens paid by the subjects (hereafter, called "contributions") are shown in Table 4, which summarizes our experimental results. The average contribution under the non-disclosed-risk condition, 5.43 (S.D. = 5.90), is higher than that under the disclosed-risk condition, 4.78 (S.D. = 3.77). The difference is statistically significant (*p*-value < 0.01). The average and standard deviation of the contributions in a period are represented in Fig 3A–3D by a dot and its corresponding vertical line, respectively. The upper left and upper right panels of Fig 3A–3D reflect the series system under the non-disclosed-risk condition and the disclosed-risk condition, respectively. Although the average contribution under the non-disclosed-risk condition gradually decreases and approaches that under the

**Table 3. Demographics.**

| | Series | | | Parallel | | |
|---|---|---|---|---|---|---|
| | Session 1 | Session 2 | Session 3 | Session 4 | Session 5 | Session 6 |
| Number of Male | 8 | 5 | 5 | 5 | 5 | 9 |
| Number of Female | 4 | 7 | 7 | 7 | 7 | 3 |
| Average payoffs (JPY) | 3904 | 3415 | 4041 | 4616 | 4603 | 4788 |
| Number of people dropped by lottery | 3 | 2 | 4 | 2 | 4 | 7 |

**Table 4. Summary of the experimental results.**

| | Non-disclosed-risk (1–30 periods) | Disclosed-risk (31–60 periods) | *p*-value of mean test: |
|---|---|---|---|
| Series system | (a) 5.43 (5.90) | (a) 4.78 (3.77) | < 0.01 *** |
| | (b) 0.31 (0.28) | (b) 0.49 (0.29) | < 0.01 *** |
| | (c) 4.06 (1.33) | (c) 3.10 (1.14) | < 0.01 *** |
| | (d) 680.19 (82.84) | (d) 819.25 (85.18) | < 0.01 *** |
| Parallel system | (a) 3.96 (4.85) | (a) 2.45 (2.86) | < 0.01 *** |
| | (b) 0.75 (0.23) | (b) 0.78 (0.18) | < 0.01 *** |
| | (c) 1.41 (0.99) | (c) 1.47 (1.05) | 0.71 |
| | (d) 1054.89 (73.82) | (d) 1093.22 (63.72) | 0.02 ** |

(a) Average contribution [tokens], (b) Average system reliability [–], (c) Average number of accidents per period, (d) Average payoff over all 30 periods, excluding the participation fee [tokens]

Values in parentheses are standard deviations.

*, **, and *** denote statistical significance at the 10%, 5%, and 1% levels, respectively.

(A)

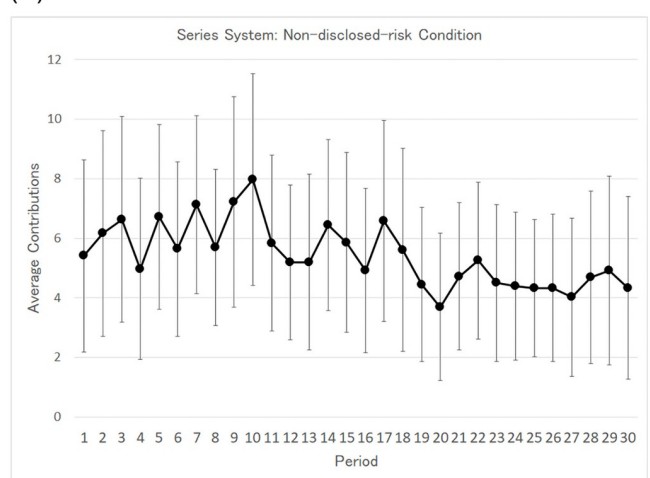

(B)

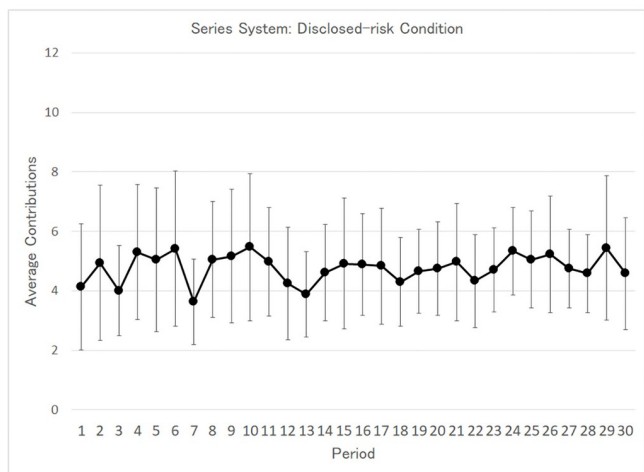

(C)

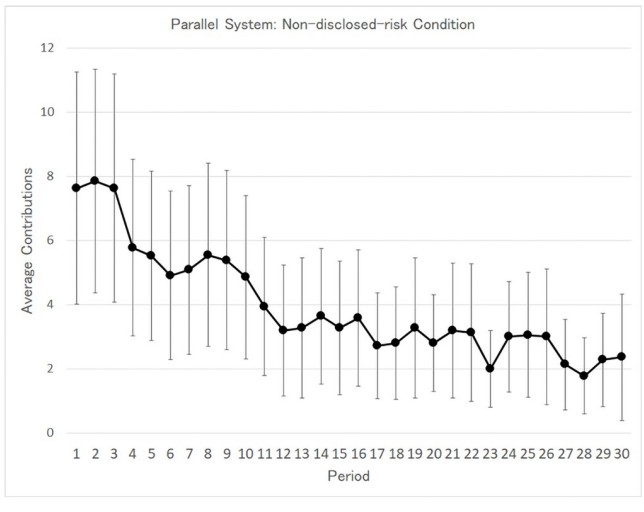

(D)

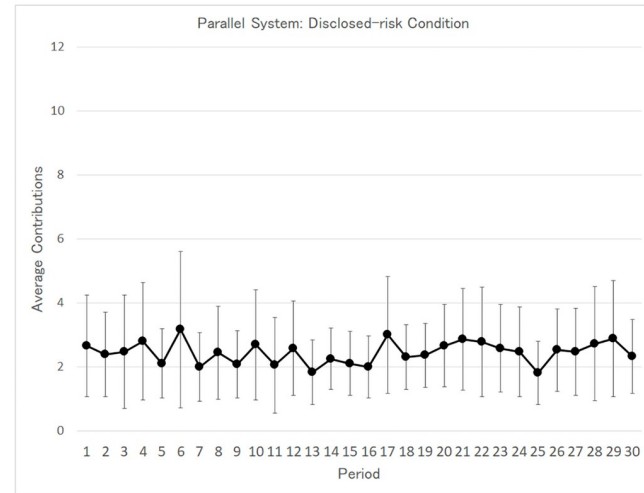

**Fig 3.** A-D. Average and standard deviation of each period's contribution.

disclosed-risk condition as the period progresses, the average contribution remains stable under the disclosed-risk condition. The standard deviations tend to be smaller under the disclosed-risk condition than under the non-disclosed-risk condition. A possible explanation for this reduction in the standard deviations is as follows. Under the disclosed-risk condition, subjects have access to information on the relationship between their contributions and the unit's reliability (Table 1). By referring to Table 1, the subjects could avoid the process of trial and error. That is, they did not need to make very large or very small contributions in order to guess and verify the relationship, as they did under the non-disclosed-risk condition. As a result, the standard deviations of contributions under the disclosed-risk condition decreased.

We conducted a statistical test of whether the average contribution significantly differs from the Nash equilibria of (0, 0) and (6, 6). Since the Nash equilibria presented in section 0 are those of the complete information model, they should be compared with the experimental results under the disclosed-risk condition. The average contribution in the last period of the disclosed-risk condition is 4.58 (S.D. = 3.71), which is not very different from the latter equilibrium of six. However, a *t*-test rejects the null hypothesis that the average contribution in the last period is six (*p*-value = 0.03), and the null that the average contribution in the last period is zero is also rejected (*p*-value < 0.01).

We estimated a regression model to examine the cause of the variation in contributions and determine whether the behavior of the subjects changed when the risk probability was disclosed. The dependent variable $cont_{i,t}$ is the number of contributions of subject $i$ (= 1, . . ., 36) in period $t$ (= 2, . . ., 60). The independent variables are $d_t$, $cont_{i,t-1}$, $cont_{i's\ partner,t-1}$, $aac_{i,t-1}$, and the cross terms that are the products of $d_t$ and each variable. $d_t$ is a dummy variable that takes a value of zero if period $t$ is part of the non-disclosed-risk condition and a value of one otherwise, $cont_{i,t-1}$ is the number of contributions of subject $i$ in period $t - 1$, $cont_{i's\ partner,t-1}$ is the number of contributions of subject $i$'s partner in period $t - 1$, $ac_{i,t-1}$ is a dummy variable that takes a value of one if subject $i$ had an accident in period $t - 1$ and a value of zero otherwise, and $aac_{i,t-1}$ is the accumulated number of accidents subject $i$ has had by period $t - 1$. The regression model is as follows:

$$cont_{i,\ t} = \alpha + \beta_0 d_t + \beta_1 cont_{i,\ t-1} + \beta_2 d_t * cont_{i,\ t-1} + \beta_3 cont_{i's\ partner,\ t-1} + \beta_4 d_t * cont_{i's\ partner,\ t-1}$$

$$+ \beta_5 ac_{i,t-1} + \beta_6 d_t * ac_{i,t-1} + \beta_7 aac_{i,\ t-1} + \beta_8 d_t * aac_{i,\ t-1} + e_{i,t} \qquad (1)$$

The coefficients on $cont_{i,t-1}$, $cont_{i's\ partner,t-1}$, $ac_{i,t-1}$, $aac_{i,t-1}$ are those for the non-disclosed-risk condition, and the coefficients on the cross terms reflect the difference in coefficients between the non-disclosed-risk condition and the disclosed-risk condition. The slopes of the disclosed-risk condition can be measured as the sums of corresponding coefficients, like, for example, $\beta_1$ and $\beta_2$. The significance of those slopes can be tested using an *F*-test.

In our experiment, the same subjects decide how many contributions to make 30 times for both the non-disclosed-risk condition and the disclosed-risk condition. Therefore, the data set has a panel (longitudinal) data structure. The Hausman test rejects a random effect model, so we present the estimation results of a fixed effect model in the left column of Table 5. For more information concerning panel data analysis, see econometrics textbooks, such as Wooldridge [15].

As presented in Table 5, the coefficient on $aac_{i,t-1}$ is -0.11 and is statistically significant. The subjects had no information about the likelihood of accident under the non-disclosed-risk condition. Therefore, it seems that they made conservatively many contributions under this condition, especially in the first periods, and they adjusted the number of contributions by observing their accident records in the experiment. In fact, the result of the regression analysis shows that the subjects decreased the number of contributions as the accumulated number of

**Table 5. Regression analysis on contributions.**

| Independent variables | Dependent variable: $cont_{i,t}$ | | | |
|---|---|---|---|---|
| | Series system | | Parallel system | |
| constant | 4.85 (0.48) | *** | 1.89 (0.22) | *** |
| $d_t$ | -1.29 (0.65) | ** | -0.26 (0.31) | |
| $cont_{i,t-1}$ | 0.31 (0.03) | *** | 0.51 (0.02) | *** |
| $d_t * cont_{i,t-1}$ | -0.06 (0.05) | | -0.15 (0.04) | *** |
| $cont_{i's\ partner,t-1}$ | 0.04 (0.02) | | 0.05 (0.02) | *** |
| $d_t * cont_{i's\ partner,t-1}$ | 0.03 (0.04) | | -0.08 (0.04) | ** |
| $ac_{i,t-1}$ | -0.32 (0.33) | | 0.87 (0.24) | *** |
| $d_t * ac_{i,t-1}$ | -0.02 (0.44) | | -0.74 (0.33) | ** |
| $aac_{i,t-1}$ | -0.11 (0.02) | *** | -0.16 (0.04) | *** |
| $d_t * aac_{i,t-1}$ | 0.10 (0.03) | *** | 0.16 (0.05) | *** |
| N | 2088 | - | 2088 | - |
| F[9, 2043] | 34.73 | *** | 97.77 | *** |
| $R^2$ (within) | 0.13 | - | 0.30 | - |

Values in parentheses are standard deviations.

*, **, and *** denote statistical significance at the 10%, 5%, and 1% levels, respectively.

accidents increased. From the subjects' point of view, contributions are investments to earn 25 tokens. The estimation result shows that subjects who lost their investments many times started to hesitate to invest in order to ensure a large payoff even in the case of an accident.

The coefficient on $d_t * acc_{i,t-1}$ is 0.10 and is statistically significant. This means that subjects actually changed their behavior when the risk probability was revealed. $\beta_7 + \beta_8$, which is the slope of $acc_{i,t-1}$ for the disclosed-risk condition, is -0.01 and is not statistically significant ($p$-value = 0.60). This result means that the subjects did not decrease their contributions regardless of the accumulated number of accidents under the disclosed-risk condition. The subjects did not need to pay attention to the accumulated number of accidents under the disclosed-risk condition because they made their decisions based on the disclosed risk probabilities of their machine units, as shown in Table 1. Aside from $acc_{i,t-1}$ and $d_t * acc_{i,t-1}$, only $cont_{i,t-1}$ is statistically significant.

**4.2.2. System reliability.** As shown in Table 4, the average system reliabilities of the non-disclosed-risk condition and the disclosed-risk condition are 0.31 (S.D. = 0.28) and 0.49 (S.D. = 0.29), respectively. The difference between them is statistically significant ($p$-value < 0.01). Note that the average system reliability of the disclosed-risk condition is higher than that of the non-disclosed-risk condition despite the fact that the average contribution is higher under the non-disclosed-risk condition than under the disclosed-risk condition. The average number of total accidents per period in first 30 periods in all 3 sessions is 4.06 (S.D. = 1.33) under the non-disclosed-risk condition, which is significantly higher than the average of 3.10 (S.D. = 1.14) under the disclosed-risk condition ($p$-value < 0.01). This finding is consistent with the result on system reliability.

In Fig 4A–4D, the average and standard deviation of the system reliabilities in a period are represented by a dot and its corresponding vertical line, respectively. The upper left and upper right panels of Fig 4A–4D show the non-disclosed-risk condition and the disclosed-risk condition, respectively. Clearly, the average system reliabilities under the disclosed-risk condition are higher than those under the non-disclosed-risk condition. In order to examine whether there is a structural change in the time series variation of the system reliability between the

(A)

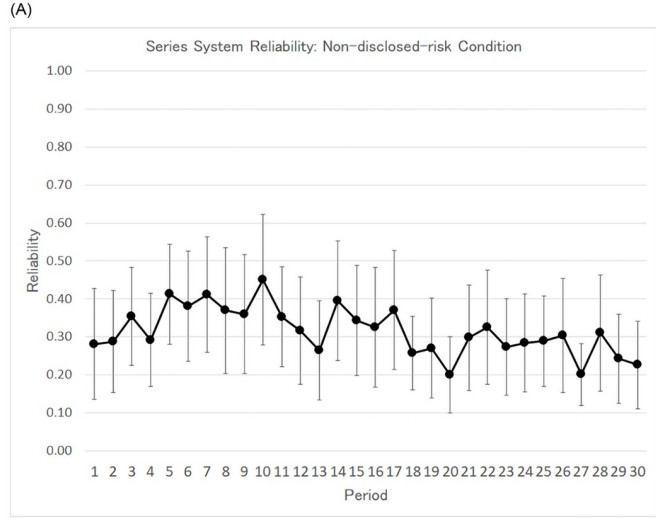

(B)

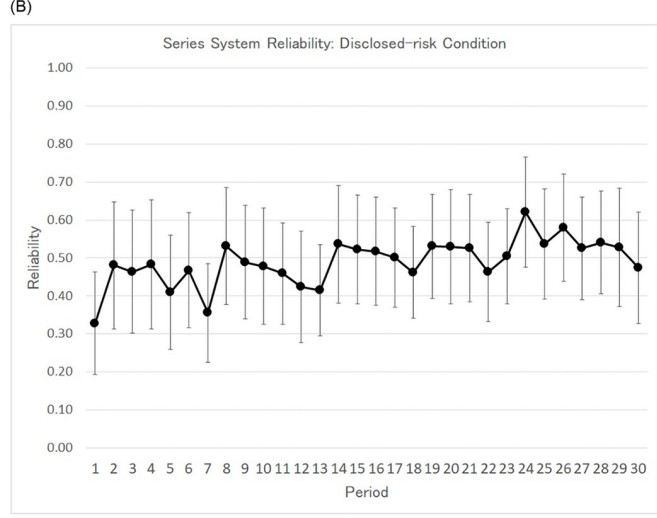

(C)

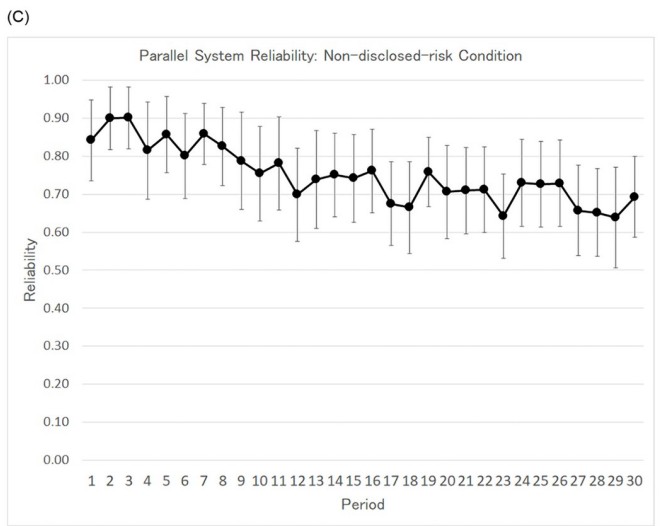

(D)

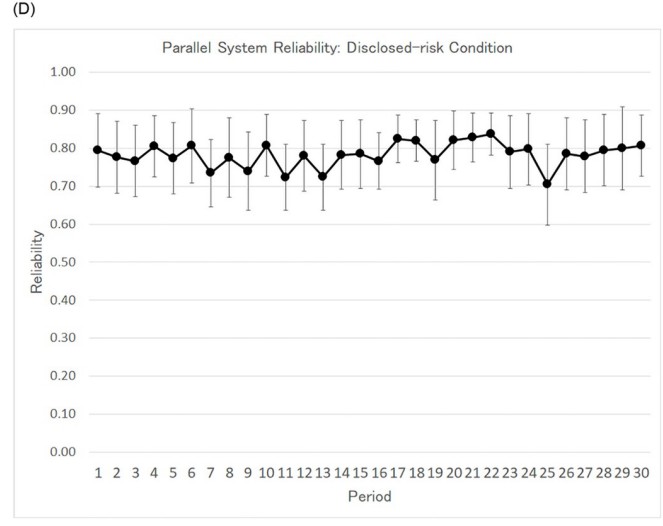

**Fig 4.** A-D. Average and standard deviation of each period's system reliability.

non-disclosed-risk and the disclosed-risk condition, we estimated the following regression model.

$$reli_{j,\,t} \; = \; \beta_0 + \beta_1 t + \beta_2 d_t + \beta_3 t * d_t + e_{j,\,t} \tag{2}$$

where $reli_{j,t}$ is the system reliability of group $j$ (= 1, . . ., 18) at period $t$ (= 1, . . ., 60) and $d_t$ is a dummy variable that takes a value of zero if period $t$ is in the non-disclosed-risk condition and a value of one otherwise. The result presented in the left column of Table 6 shows statistical evidence that there is a structural change. Not only is the coefficient on $t$ (-0.004) negatively significant while that on $t * d_t$ (0.008) is positively significant, but also their sum (0.004) is significantly positive (for all, p < 0.01). This result means that the system reliability significantly decreases under the non-disclosed-risk condition and increases under the disclosed-risk condition as the period progresses.

**Table 6. Regression analysis on system reliabilities.**

| Independent variable | Dependent variable: $reli_{j,t}$ | | | |
|---|---|---|---|---|
| | Series system | | Parallel system | |
| constant | 0.38 (0.02) | *** | 0.86 (0.01) | *** |
| $t$ | -0.004 (0.001) | *** | -0.007 (0.001) | *** |
| $d_t$ | -0.08 (0.05) | | -0.12 (0.04) | *** |
| $t^*d_t$ | 0.008 (0.00) | *** | 0.008 (0.00) | *** |
| $N$ | 2160 | - | 2160 | - |
| $F[3, 2156]$ | 79.74 | *** | 39.41 | *** |
| Adj. $R^2$ | 0.099 | - | 0.051 | - |

Values in parentheses are standard deviations.

*, **, and *** denote statistical significance at the 10%, 5%, and 1% levels, respectively.

The average payoff under the disclosed-risk condition is 819.25 (S.D. = 85.18) and is higher than that under the non-disclosed-risk condition, 680.19 (S.D. = 82.84). The difference is statistically significant ($p$-value < 0.01). This result reflects the fact that under the disclosed-risk condition, the subjects had fewer accidents but made lower contributions.

## 4.3 Parallel system

**4.3.1 Contributions.** Table 4 shows average contribution under the non-disclosed-risk condition, 3.96 (S.D. = 4.85), is higher than that under the disclosed-risk condition, 2.45 (S.D. = 2.86). The difference is statistically significant ($p$-value < 0.01). The lower left and lower right panels of Fig 3A–3D show the average contributions under the non-disclosed-risk condition and the disclosed-risk condition for the parallel system. As is the case with the series system, although the average contribution of the non-disclosed-risk condition decreases and approaches that of the disclosed-risk condition as the period progresses, the average contribution of the disclosed-risk condition remains stable. The result for the standard deviations is similar to that of the series system. The standard deviations of contributions under the disclosed-risk condition are smaller than those under the non-disclosed-risk condition.

The Nash equilibria of the parallel system are (0, 6), (6, 0), and (3, 3), as discussed in section 0. We form the null hypothesis that the average contribution of the last period of the disclosed-risk condition is three, since the average contribution in the Nash equilibrium is three for each equilibrium. The average contribution of the last period of the disclosed-risk condition is 2.33 (S.D. = 2.29). The $t$-test result shows that the null hypothesis is not rejected at the 5% significance level, but it is rejected at the 10% significance level ($p$-value = 0.09).

The right column of Table 5 shows the results for the fixed effect model of panel regression (1), whose dependent variable is $cont_{i,t}$. The coefficients on the variables corresponding to the non-disclosed-risk condition, $cont_{i,t-1}$, $cont_{i's\ partner,t-1}$, $ac_{i,t-1}$, and $aac_{i,t-1}$, are statistically significant. This result is more striking in the regression analysis of the parallel system than in that of the series system. Indeed, for the series system, only the variables relevant to the accumulated number of accidents are significant, with the exception of $cont_{i,t-1}$. Since the subjects could not see Table 1 under the non-disclosed-risk condition, they adjusted their levels of contributions using all of the information they could obtain in the experiment as a reference. Furthermore, all of the coefficients on the cross terms are significant, which means that the subjects actually changed their behavior when the risk probability was revealed under the disclosed-risk condition. None of the coefficients on the variables corresponding to the disclosed-

risk condition are significant except for $\beta_{1+}\beta_2$. This result means that, under the disclosed-risk condition, the subjects did not take into account information on their partner's contribution and accident in period $t-1$ and the accumulated number of accidents by period $t-1$. Instead, they used only Table 1 to make their contribution decisions.

**4.3.2 System reliability.** As shown in Table 4, the average system reliabilities of the non-disclosed-risk condition and the disclosed-risk condition are 0.75 (S.D. = 0.23) and 0.78 (S.D. = 0.18), respectively. This difference is statistically significant ($p$-value < 0.01). As is the case with the series system, the average system reliability of the disclosed-risk condition is higher than that of the non-disclosed-risk condition despite the fact that the average contribution is higher under the non-disclosed-risk condition than under the disclosed-risk condition.

The lower left and lower right panels of Fig 4A–4D shows the averages and standard deviations of the system reliabilities under the non-disclosed-risk and the disclosed-risk conditions, respectively. The right column of Table 6 presents the estimated coefficients of regression model (2). As with the results for the series system, the statistical evidence shows there was a structural change in the coefficient on $t$ when the risk probability was revealed under the disclosed-risk condition. The coefficient on $t$ (-0.007) is negatively significant, and that on $t^*d_t$ (0.008) is positively significant (for both, the $p$-value < 0.01). However, their sum (0.001) is not significantly different from zero ($p$-value = 0.26), which means that the system reliability of the disclosed-risk condition remains stable.

The average number of accidents is 1.41 (S.D. = 0.99) under the non-disclosed-risk condition, which is not significantly different from 1.47 (S.D. = 1.05) under the disclosed-risk condition ($p$-value = 0.71). This result reflects the fact that the difference in the system reliability between the non-disclosed-risk condition and the disclosed-risk condition is not very large even though it is statistically significant.

The average payoff under the disclosed-risk condition is 1093.22 (S.D. = 63.72), which is higher than that under the non-disclosed-risk condition, 1054.89 (S.D. = 73.82). This difference is statistically significant ($p$-value = 0.02). This result reflects the fact that subjects exerted less effort under the disclosed-risk condition than under the non-disclosed-risk condition, whereas the numbers of accidents per period did not differ substantially across the two conditions.

## 5. Discussion

### 5.1 Effect of disclosing the relation between effort and unit reliability on system reliability

First, we discuss the effect of disclosing the relation between effort and unit reliability on system reliability based on the last 10 periods of the non-disclosed-risk condition and all periods of the disclosed-risk condition. The subjects used a process of trial and error to learn how to determine their contributions during the first few periods of the non-disclosed-risk condition. The issue of learning is extensively discussed in the literature. For instance, Hausken and Ortmann (2008), who conducted the prisoner's dilemma, where free riding plays a role and Hausken, Banuri, Gupta, and Abbink (2015) who conducted a type of coordination game for terrorist attacks [16, 17].

We treat the last 10 periods of the non-disclosed-risk condition as considered to have been in a steady state. The average contribution of the non-disclosed-risk condition gradually converges to that of the disclosed-risk condition for both systems, as mentioned in sections 4.2.1 and 4.3.1. For the series system, the average contribution in the last 10 periods of the non-disclosed-risk condition is 4.55 (S.D. = 5.33), and there is no statistically significant difference between this value and the average contribution under the disclosed-risk condition of 4.78

(S.D. = 3.77) ($p$-value = 0.46). For the parallel system, the same result holds, with the average contributions being 2.59 (S.D. = 3.52) and 2.45 (S.D. = 2.86) for the last 10 periods of the non-disclosed-risk condition and the disclosed-risk condition, respectively ($p$-value = 0.48). This result indicates that subjects tried to find the optimal level of contributions by observing their own and their counterpart's numbers of contributions and the occurrence of accidents. However, this convergence of the averages does not mean that the subjects' behavior in the last 10 periods of the non-disclosed-risk condition is precisely the same as that under the disclosed-risk condition, as indicated by the standard deviations of the contributions under both conditions. The standard deviation of the contributions in the last 10 periods of the non-disclosed-risk condition is larger than that of the disclosed-risk condition for both the series and the parallel systems.

The average system reliability in the last 10 periods of the non-disclosed-risk condition is 0.28 for the series system, which is significantly lower than that of the disclosed-risk condition of 0.49 ($p$-value < 0.01). As for the parallel system, the same result holds, with a system reliability of 0.69 in the last 10 periods of the non-disclosed-risk condition and 0.78 under the disclosed-risk condition ($p$-value < 0.01). Even though the average contribution in the last 10 periods of the non-disclosed-risk condition is not different from that of the disclosed-risk condition, the average system reliability is significantly higher under the disclosed-risk condition than in the last 10 periods of the non-disclosed-risk condition for both systems.

When we incorporate the data from the first 20 periods of the non-disclosed-risk condition, where subjects' contributions were relatively higher, the outline of the result does not change. As shown in Table 4, the average contribution under the non-disclosed-risk condition is higher than that under the disclosed-risk condition for both the series and the parallel system. As discussed in sections 4.2.2 and 4.3.2, the average system reliability of the disclosed-risk condition is significantly higher than that of the non-disclosed-risk condition for both systems. This result is interesting because it contradicts the result that the subjects actually made more contributions to maintain the machine units under the non-disclosed-risk condition than under the disclosed-risk condition.

This contradiction is related to the fact that the variance of the contributions under the non-disclosed-risk condition is larger than that under the disclosed-risk condition, which means that the contributions of some subjects are relatively low in the former condition. These low contributions result in low system reliability. This effect is noticeable in the series system, which can be easily affected by just one subject's contribution, but it is also observed in the parallel system.

This result offers a new perspective on risk assessment and the disclosure of risk probability. That is, the disclosure of risk probability could coordinate people's actions to reduce risk. Since the subjects could refer to Table 1 and share common information under the disclosed-risk condition of our experiment, their contributions were concentrated around the Nash equilibrium, and few subjects made extremely high or low contributions. They succeeded in simultaneously reducing their maintenance efforts and improving the system reliability. As a result, the average payoff that subjects received was significantly higher under the disclosed-risk condition than under the non-disclosed-risk condition for both the series and the parallel system (Table 4). This finding means that the economic welfare of subjects improved by disclosing the accident risk probability.

There are several Nash equilibria for the series system such that (0, 0) and (6, 6) imply that both subjects spend nothing or less than half of their efforts. In the non-disclosure condition, during the last 10 periods, the contributions are less than 5 so that system liability reduces. On the other hand, in the disclosure condition, during the last 10 periods, their efforts spent are

more than 5 so that the system liability increases. The disclosure condition makes the contributions close to the Nash equilibrium (6, 6) which is good for the system liability.

For the parallel system, there are several Nash equilibria such that (6, 0), (0, 6) and (3, 3) imply that one of them spend efforts or both of them spend a small amount of their efforts. In the non-disclosure condition, during the last 10 periods, one of them spent around 3 but the other do free ridings so that system liability reduces. On the other hand, in the disclosure condition, both of them spent around three so that system liability induces. The disclosure condition makes the contributions close to the Nash equilibrium (3, 3) which is good for the system liability.

In the multiple Nash equilibria, since the mixed strategy will be taken, the system liability will not be robust. Disclosing the relation between effort and unit reliability has an effect to their contribution or behavior to one Nash equilibria. Such a change from the other equilibrium to one equilibrium is affected by the subjects' behaviors. Next section we show the difference of their behaviors to induces the different Nash equilibria.

## 5.2 Effect of disclosing the relation between effort and unit reliability on subjects' behavior

As shown in Table 5, the estimation results of regression model (1) indicate that the subjects actually changed their behavior when the risk probability, that is, the information in Table 1, was revealed under the disclosed-risk condition. In particular, we focus on the result that the subjects did not take into account the information on the partner's contribution and accident in period $t − 1$ and the accumulated number of accidents by period $t − 1$ under the disclosed-risk condition. Since, under the disclosed-risk condition, subjects could obtain information on the risk probability from Table 1, they determined how many contributions to make based on only that risk probability without considering other factors. In other words, the subjects' decision-making process was simplified, or its psychological cost was reduced. Under the disclosed-risk condition, subjects could reduce not only their contributions but also the cost of decision-making. This is another positive effect of the disclosure of risk probability.

This effect is analogous to that of hazard maps and durable life information in food production. Both provide information on safety and help to avoid danger. If people have hazard maps, they do not have to look for safe places by themselves when disasters happen. If they know durable lives of foods, they do not have to examine the safety and hygiene of the foods themselves when eating. This kind of information on safety can simplify the decision-making process and reduce its cost.

## 6. Conclusions

We conducted an economic experiment on the basis of Hausken's (2002) theoretical model that merged PRA and game theory with a particular focus on the effect of disclosing the relation between effort and unit reliability on the subjects' behavior and system reliability. Our experimental results show that disclosing the relation between effort and unit reliability has two positive effects. First, subjects succeeded in improving the system reliability while reducing their efforts to reduce the risk of their units when the risk probability was disclosed. Second, disclosing the risk probability simplified the subjects' decision-making process and reduced its cost. We believe that this study is the first step toward experimentally examining the integrated model of PRA and game theory, and the results offer a new perspective on the function of the disclosure of risk probability, namely, that it has the potential to coordinate people's actions to reduce risk.

The following two future research agendas can be made. The first one concerns the heterogeneous risks of the machine units. In this study, each player faced the same failure risk, as shown in Table 1. However, in the real industrial setting, there is a mix of equipment with different installation times, i.e., different degrees of deterioration. The relationship between maintenance effort and failure probability is considered to be different for newer and older equipment. In general, the older the equipment, the more effort would be required to reduce the failure probability. The effect of such heterogeneous risks on each player's behavior and on system reliability in equilibrium should be examined. Second one is related to the positive effect of risk probability disclosure discussed in section 5.2. In this study, we provided an example in which risk probability disclosure helped subjects improve system reliability and avoid danger. However, there could be circumstances where this disclosure brings about negative outcomes. For instance, people who believe the information of a hazard map and take refuge where instructed may end up suffering damage from a disaster such as a tsunami if the tsunami hits the place of refuge. In this case, the hazard map results in heavy damage to the population by gathering everyone in one place. Our future experimental research will be focused on the conditions under which risk information disclosure produces negative outcomes.

## Supporting information

**S1 File.**
(XLSX)

## Author Contributions

**Conceptualization:** Ryoji Makino, Kenju Akai, Keiko Aoki.

**Data curation:** Ryoji Makino, Kenju Akai, Takanori Kudo, Keiko Aoki.

**Formal analysis:** Ryoji Makino, Kenju Akai, Jun-ichi Takeshita.

**Funding acquisition:** Ryoji Makino.

**Investigation:** Ryoji Makino, Kenju Akai, Jun-ichi Takeshita, Takanori Kudo, Keiko Aoki.

**Methodology:** Ryoji Makino, Kenju Akai, Jun-ichi Takeshita, Takanori Kudo.

**Project administration:** Ryoji Makino, Kenju Akai.

**Resources:** Ryoji Makino, Kenju Akai.

**Supervision:** Ryoji Makino, Kenju Akai.

**Validation:** Ryoji Makino, Kenju Akai.

**Visualization:** Ryoji Makino, Kenju Akai.

**Writing – original draft:** Ryoji Makino, Kenju Akai, Keiko Aoki.

**Writing – review & editing:** Ryoji Makino, Kenju Akai, Keiko Aoki.

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
