## [Decision Letter · Decision Letter 0]

15 Dec 2020

PONE-D-20-31656

Effect of Disclosing the Relation between Effort and Unit Reliability on System Reliability: An Economic Experiment

PLOS ONE

Dear Dr. Akai,

Thank you for submitting your manuscript to PLOS ONE. After careful consideration, we feel that it has merit but does not fully meet PLOS ONE’s publication criteria as it currently stands. Therefore, we invite you to submit a revised version of the manuscript that addresses the points raised during the review process.

We look forward to receiving your revised manuscript.

Kind regards,

Luo-Luo Jiang, Ph.D.

Academic Editor

PLOS ONE

Journal Requirements:

2.) We suggest you thoroughly copyedit your manuscript for language usage, spelling, and grammar. If you do not know anyone who can help you do this, you may wish to consider employing a professional scientific editing service.  

3.) We note that you have indicated that data from this study are available upon request. PLOS only allows data to be available upon request if there are legal or ethical restrictions on sharing data publicly. For information on unacceptable data access restrictions, please see http://journals.plos.org/plosone/s/data-availability#loc-unacceptable-data-access-restrictions.

Reviewers' comments:

Reviewer's Responses to Questions

**Comments to the Author**

1. Is the manuscript technically sound, and do the data support the conclusions?

Reviewer #1: Partly

Reviewer #2: Yes

2. Has the statistical analysis been performed appropriately and rigorously? 

Reviewer #1: Yes

Reviewer #2: Yes

3. Have the authors made all data underlying the findings in their manuscript fully available?

Reviewer #1: No

Reviewer #2: No

4. Is the manuscript presented in an intelligible fashion and written in standard English?

Reviewer #1: Yes

Reviewer #2: Yes

5. Review Comments to the Author

Reviewer #1: In this paper, a behavioral experiment is studied, which discloses the relation between effort, rish probability and unit reliability. The results of this experiment are enlightening and worthy of publication.

The only problem is that we hope that the authors can attach the detailed experimental operation process and statistical table to the paper for comparison and reproduction.

What's more, in the experiment, are there any new effects and changes in the results if heterogeneous risks are used?

The figures in the article are very vague, so it is suggested to make them clear.

There are many typos, such as “Error! Reference source not found.” in this paper.

Reviewer #2: This paper tries to explore the effect of disclosing risk on human decision making, which is definitely an interesting topic. Based on the integrated models of PRA and game theory, which is initiated by Hausken(2002), the authors developed their experiments to find insights. However, what is the contribution of this paper on Hausken(2002) is unclear. In other words, the motivation of this work should be highlighted.

6. PLOS authors have the option to publish the peer review history of their article (what does this mean?). If published, this will include your full peer review and any attached files.

Reviewer #1: No

Reviewer #2: No

---

## [Author Response · Author response to Decision Letter 0]

31 Jan 2021

Reviewer #1: In this paper, a behavioral experiment is studied, which discloses the relation between effort, rish probability and unit reliability. The results of this experiment are enlightening and worthy of publication.

The only problem is that we hope that the authors can attach the detailed experimental operation process and statistical table to the paper for comparison and reproduction.

Reply) We add the all data in the supplemental material. We show the detailed experimental operation steps by the blue colored sentences in the design and procedure section. We show the subject tables. The statistical results are shown in Table 3.

What's more, in the experiment, are there any new effects and changes in the results if heterogeneous risks are used?

Reply) Line 29-30, 37-38, 210-250, 257-258, and 526-534. Hausken does not consider the heterogeneous risk model. This is very important model when we consider the real world. We cannot discuss the result of heterogeneous model. We add the notification and future issue about this problem by the blue colored sentences in the conclusion section. This is our first priority in the future project.

The figures in the article are very vague, so it is suggested to make them clear.

Reply) We modify the figures.

There are many typos, such as “Error! Reference source not found.” in this paper.

Reply) They are captions of Figures. We modified them.

Reviewer #2: This paper tries to explore the effect of disclosing risk on human decision making, which is definitely an interesting topic. Based on the integrated models of PRA and game theory, which is initiated by Hausken(2002), the authors developed their experiments to find insights. However, what is the contribution of this paper on Hausken(2002) is unclear. In other words, the motivation of this work should be highlighted.

Reply)　Line 39-47, 51-52, 54-58, 169-170, and 189-200. We clarify the motivation and contribution of this study in red colored sentences in the introduction section. Hausken only consider the theoretical equilibrium in one-shot game under the complete information condition. Our study conducts laboratory experiments to evaluate whether the game achieves the equilibrium by actual human subjects. We also extend this model into repeated games to test find what happens after they experience the accident and know the counterpart’s effort as shown in Tables 4 and 5.

---

## [Decision Letter · Decision Letter 1]

24 Mar 2021

Effect of Disclosing the Relation between Effort and Unit Reliability on System Reliability: An Economic Experiment

PONE-D-20-31656R1

Dear Dr. Akai,

We’re pleased to inform you that your manuscript has been judged scientifically suitable for publication and will be formally accepted for publication once it meets all outstanding technical requirements.

Kind regards,

Camelia Delcea

Academic Editor

PLOS ONE

Additional Editor Comments (optional):

Reviewers' comments:

Reviewer's Responses to Questions

**Comments to the Author**

1. If the authors have adequately addressed your comments raised in a previous round of review and you feel that this manuscript is now acceptable for publication, you may indicate that here to bypass the “Comments to the Author” section, enter your conflict of interest statement in the “Confidential to Editor” section, and submit your "Accept" recommendation.

Reviewer #1: All comments have been addressed

2. Is the manuscript technically sound, and do the data support the conclusions?

Reviewer #1: Yes

3. Has the statistical analysis been performed appropriately and rigorously? 

Reviewer #1: Yes

4. Have the authors made all data underlying the findings in their manuscript fully available?

Reviewer #1: Yes

5. Is the manuscript presented in an intelligible fashion and written in standard English?

Reviewer #1: Yes

6. Review Comments to the Author

Reviewer #1: The authors have answered all of my questions. I would like to recommend accepting this paper at this moment.

7. PLOS authors have the option to publish the peer review history of their article (what does this mean?). If published, this will include your full peer review and any attached files.

Reviewer #1: No

---

## [Editor Report · Acceptance letter]

29 Mar 2021

PONE-D-20-31656R1 

Effect of Disclosing the Relation between Effort and Unit Reliability on System Reliability: An Economic Experiment 

Dear Dr. Akai:

I'm pleased to inform you that your manuscript has been deemed suitable for publication in PLOS ONE. Congratulations! Your manuscript is now with our production department. 

Kind regards, 

on behalf of

Dr. Camelia Delcea 

Academic Editor

PLOS ONE